# Systemic Signals Induced by Single and Combined Abiotic Stimuli in Common Bean Plants

**DOI:** 10.3390/plants12040924

**Published:** 2023-02-17

**Authors:** Ádrya Vanessa Lira Costa, Thiago Francisco de Carvalho Oliveira, Douglas Antônio Posso, Gabriela Niemeyer Reissig, André Geremia Parise, Willian Silva Barros, Gustavo Maia Souza

**Affiliations:** 1Laboratory of Plant Cognition and Electrophysiology, Department of Botany, Institute of Biology, Federal University of Pelotas, Capão do Leão CEP 96160-000, Rio Grande do Sul, Brazil; 2School of Biological Sciences, University of Reading, Reading RG6 6AH, UK

**Keywords:** electrome, heat shock, time dispersion analysis, turgor pressure, wound, plant attention

## Abstract

To survive in a dynamic environment growing fixed to the ground, plants have developed mechanisms for monitoring and perceiving the environment. When a stimulus is perceived, a series of signals are induced and can propagate away from the stimulated site. Three distinct types of systemic signaling exist, i.e., (i) electrical, (ii) hydraulic, and (iii) chemical, which differ not only in their nature but also in their propagation speed. Naturally, plants suffer influences from two or more stimuli (biotic and/or abiotic). Stimuli combination can promote the activation of new signaling mechanisms that are explicitly activated, as well as the emergence of a new response. This study evaluated the behavior of electrical (electrome) and hydraulic signals after applying simple and combined stimuli in common bean plants. We used simple and mixed stimuli applications to identify biochemical responses and extract information from the electrical and hydraulic patterns. Time series analysis, comparing the conditions before and after the stimuli and the oxidative responses at local and systemic levels, detected changes in electrome and hydraulic signal profiles. Changes in electrome are different between types of stimulation, including their combination, and systemic changes in hydraulic and oxidative dynamics accompany these electrical signals.

## 1. Introduction

As sessile organisms, plants cannot move away from threats. To circumvent these conditions, they have developed monitoring systems, signaling mechanisms, and responses to various stimuli enabling their survival, development, and reproduction [1].

Sometimes, a stimulus on the plant can occur at a local level (e.g., cell or tissue) and trigger responses in regions far away from the affected site (i.e., systemic response). The local and systemic level of responses can be induced by different signals that propagate over long distances in plants [2,3]. This systemic signaling aspect relies on different traveling molecules, such as phytohormones, small peptides, micro RNAs, calcium waves, reactive oxygen species (ROS), volatile organic compounds, and hydraulic and electrical signals (see reviews [4,5]). These signals can travel from one cell to another via apoplast or symplast pathways and by vascular tissues throughout the plant [5,6].

Among all the long-distance signals, hydraulic and electrical ones have higher propagation speeds [5,7], enabling, in certain cases, prompt decoding of the stimulus and assisting in the coordination of appropriate responses [8]. 

Plants exhibit electrical signals and spontaneous, non-evoked electrical activity associated with basic physiological processes [9] that can be altered as a result of internal or external stimulation. This stimulation leads to ion imbalances across plasma membranes, leading to transient voltage variations promoted by ion flux through ion channels and electrogenic pumps [5,7]. 

Plant electrophysiological studies have described four main types of electrical signals: action potentials (AP), variation potentials (VP), local potentials (LP), and systemic potentials (SP) (see review by [10]). However, most investigations focus on a single signal type in isolation, mostly AP or VP, considering parameters such as frequency, amplitude, the distance of propagation, and time frame. However, quite often, mixed electrical potential waves are recorded, for instance, as a result of overlapping APs and VPs (among other voltage variations), creating a complex bioelectrical information network in which several electrical signals may be superposed [4]. As a result, the term “electrome” was coined to designate the totality of ionic currents of any living entity, evoked naturally or spontaneously, recorded by non-polarized Ag/AgCl electrodes [11], and more recently, the term “plant electrome” was employed to refer to the emergent complexity of bioelectrical activity in plants [12,13]. 

Recent studies analyzing plant electrome [1,13,14,15,16,17] revealed its potential as a tool for early stress diagnosis in plants since the electrome exhibits specific patterns of responses that function as a classifiable electrical signature by standard time series analysis and machine learning techniques [15,16,17,18,19].

Hydraulics also play an important role in the network of systemic signaling in plants. The hydraulic signal results from changes in water potential (Ψw), which in turn is propagated through the hydraulic connections of the xylem, enabling the hydraulic signal to transmit over long distances in the form of pressure waves [20]. The hydraulic wave can be fast, traveling nearly at the speed of sound [21], but its relevance relies on how significant the change in turgor pressure within the cell is. The pressure changes originate in the xylem vessels and, due to low axial resistance, can be propagated rapidly to surrounding cells [22] and throughout the plant. However, dead xylem cells did not perceive pressure changes, so they must be decoded by adjacent parenchyma cells, containing a series of sensors capable of detecting turgor-induced mechanical forces [20]. Among the turgor sensors, there are the mechanosensitive channels such as MscS and MscL (small/large conductance mechanosensitive channels), TPK, MCAs [23], as well as kinase receptors such as AHK1 [23] involved in the cell wall integrity (CWI) pathway, critical during pollen germination [24]. The perception of the hydraulic signal by putative sensors leads to the conversion of the physical signal into a chemical one, which will mediate various adaptive responses [20]. Although it is not known how hydraulic waves are linked to electrical, Ca^2+^, and ROS signals, it has been proposed that mechanosensitive channels permeable to Ca^2+^ and other cations and anions could detect systemic hydraulic waves in tissues distant from the stress site and convert them into Ca^2+^ signals [6,10]. 

Even though much of the research evaluates the signals separately, studies point out that the integration of these signals is necessary for a more efficient response to stimuli [5,10,24,25]. 

Signal integration is based on the activation of various membrane proteins that mediate different signaling pathways. Electrical signals and Ca^2+^ waves, which occur during systemic acquired resistance (SAR), depend on Ca^2+^-permeable glutamate receptor (GLR) type channels, and Ca^2+^ and ROS waves that occur during systemic acquired acclimation (SAA) are associated with the function of the respiratory burst oxidase homologous protein D (RBOHD), involved in ROS generation [26]. 

Although our knowledge of how stresses affect plants when applied individually is vast, in nature, plants often encounter more than one environmental stimulation/stress condition at a time, resulting in a condition termed “stress combination” [27], and little is known about how plants acclimate to a combination of different stresses, let alone how the different signals interact. The occurrence of two or more different stresses (either biotic and/or abiotic) can be challenging for plants. This challenge can be solved by additive, subtractive, and/or combinatorial effects of different pathways, networks, and mechanisms that are activated by each of the different stresses, or by triggering new responses that are specifically activated during stress combination, promoting emergent responses [28]. 

Evaluating how the integration of these signals in crop plants behaves when single and combined stimuli are applied may enable a more realistic representation of the response of these plants. Therefore, considering that local stimulation can induce systemic signaling integrating different types of signals and stress-specific responses, we investigated whether local application of single and combined abiotic stimuli induces specific changes in plant electrome and hydraulic dynamics that are able to propagate over long distances in common bean (*Phaseolus vulgaris* L.) plants.

To test this hypothesis, we (1) evaluated the dynamics of systemic signals induced by simple stimuli, followed by (2) the evaluation of the dynamics of systemic signals induced by combined stimuli. Changes in the profile of bioelectrical (electrome) and hydraulic (turgor pressure variation) signals were analyzed with techniques for time series analysis, comparing conditions before and after different stimulation conditions, as well as oxidative responses at a local and systemic level.

## 2. Results

### 2.1. Local and Systemic ROS Responses

The results of hydrogen peroxide responses (Figure 1A) showed an interaction between the analyzed factors (tissue and treatment), meaning that the effect on tissues was dependent on the treatments. The highest values, compared with the control plants, were found in the wound and thermal shock treatments, both in local and systemic tissues. However, it was verified that with the combined stresses (W+HS), hydrogen peroxide content was similar to the non-stimulated control plants.

Regarding the superoxide anion content (Figure 1B), there was no interaction between the factors, with the highest values observed in the thermal shock and wound treatments, followed by control plants, and the lowest values were found in the combined treatment (W+HS). Concerning the comparison between local and systemic tissues, there was no significant difference in any treatment.

Lipid peroxidation content (Figure 1C) did not differ between local and systemic tissues, regardless of the treatments. However, the lowest lipid peroxidation was observed in the combined treatment (W+HS) (*p* < 0.05).

### 2.2. Electrome Dynamics

#### 2.2.1. Visual Analysis

Through the visual analysis of the original time series, it was possible to detect qualitative changes in the electrome when comparing the measurements before and after the stimuli. In the series after the application of W, HS, and W + HS stimuli, the spikes (higher amplitude voltage variations, typically above 50 µV) exhibited evident changes after the stimuli in local and systemic leaves (Figure 2). In systemic leaves, the amplitude of the spikes was smaller than in local leaves. Moreover, the observations indicated that in the first minutes after stimulation, the electrome was more affected than the rest of the series.

#### 2.2.2. ApEn

Figure 3 shows the results from ApEn analysis for the W, HS, and W+HS essays comparing before and after stimulation in the local leaf. It is possible to observe that in the first moments (typically within the first 10 min), there is a significant change in the ApEn values. On the other hand, as expected, no changes were found in the control plants. On average, there was a decrease of 0.3 in the ApEn values in all treatments. However, after approximately 10 min, all values tended to match the range observed in non-stimulated control plants.

Interestingly, although a significant difference before and after stimulation in the non-stimulated plants was not observed, the control for W+HS indicated a difference of around 0.1 in complexity in the first minutes.

The ApEn results for the systemic leaves are shown in Figure 4, and as observed in the local ones, the stimuli change the entropy of the signal, with the difference being that these changes are presented in a smaller amplitude than that observed for the local leaves. A difference of 0.1 in complexity was found in the plants subjected to wounding. However, after the first few minutes, the complexity increased and remained higher than before. Plants stimulated with heat shock showed a 0.3 decrease in complexity at the beginning of the runs and returned to pre-stimulation values after around 10 min. There was a 0.2 decrease in complexity after the application of the combined stimulus, but unlike the isolated stimuli, a biphasic behavior was observed, where an increase in complexity in the first minutes was noted, followed by a reduction around 10 min later. In the second phase, the complexity increases again until it returns to the values found before the stimulus around 15 min later. 

Unexpectedly, an increase in the complexity (higher ApEn values) during the first minutes was noticed in the control plants (not stimulated) of the HS and W+HS essays.

#### 2.2.3. DFA

The results from the DFA analysis showed an increase in the correlation right after the stimuli (Figure 5). However, this value returns to a pre-stimulated condition before the first 10 min. As expected, no remarkable difference was observed in non-simulated plants through time.

The results of the DFA analysis for the systemic leaves found in Figure 6 supported the changes found by the ApEn analysis (Figure 4). However, differences in the duration of the changes observed in the electrome after combined stimulation indicated that at the systemic level, this stimulus lasted longer. No specific changes were observed in non-stimulated plants.

#### 2.2.4. Average Band Power (ABP)

The stimulus induced an increase in the energy of the low frequencies (0–0.5 and delta 0.5–4 Hz), but this increase returned to the previous values after the first 6 min (Figure 7). However, for W+HS (Figure 7c), changes in frequency energy were observed in two moments: first, an increase in the energy of the frequency was observed at the beginning of the runs, which was maintained for up to 10 min, followed by a reduction in the energy; second, the energy increased again around 50 min later and then decreased once more. The differences in the controls found in the first moments are smaller than the global average; therefore, we do not consider that there is a remarkable difference in these cases. However, the control for W+HS (Figure 7f) maintained a steady increase for more than 16 min.

In Figure 8, the results for low and delta ABP for the systemic leaves indicate that there was a change for the plants that suffer the injuries, but no substantial change was found for the control plants.

### 2.3. Measures of Turgor Pressure Variation

Measurements of turgor pressure variation indicated that after applying stimuli to local leaves, slight changes at a systemic level can be observed. In each stimulus, it was possible to observe some specific changes in the turgor pressure variation (Figure 9). Just after heat shock only, a sharper oscillation can be observed, followed by an average increase in the coefficient of variation (18.84 to 27.06%), indicating higher irregularity of the turgor pressure dynamics (Figure 9b). After the wound alone, a specific change was not clear, except for a damping in the turgor pressure oscillations approx. 15 min after the stimulus (Figure 9c). The main alterations were observed in combined stimuli (W+HS), which induced a quick and sharp increase in the turgor pressure after approx. 4 min. Moreover, the average coefficient of variation increased from 9.89% to 28.27%.

Interestingly, although no specific changes were observed in control plants as expected, the average CV% decreased (21.5–14%) after the moment when the other plants were stimulated (Figure 9a). Considering that both plants (control and stimulated) were close to each other in the Faraday cage during the electrome measurements, it would be plausible to consider a likely secondary effect caused by emitted VOCs (volatile organic compounds) from stimulated to non-stimulated plants [29].

## 3. Discussion

Environmental factors can affect plants in a spatially heterogeneous way; thus, long-distance signaling systems play an important role in the emergence of plant adaptations, allowing local responses to propagate throughout all parts of the plant body [1,30,31]. 

In addition, each stimulus can affect the plant in a specific way and trigger different responses in each module [3]. Therefore, the focus of recent research has been understanding how plants respond to different stimuli, which signals are activated, and if there is a pattern that can be identified [15,16,17,18,27]. 

Stress combination is a term used to describe the situation in which a plant is subjected to two or more abiotic stresses simultaneously. Although the combination of stress has been recognized as one of the main causes of crop loss worldwide [28,32], only more recently has this been addressed in laboratory studies at the molecular level [33,34], while there are few studies that evaluate the dynamics of signals after/during the occurrence of combined stimuli [35].

Given the complexity of the systemic signals involved, particularly the plant electrome, in this work, by applying time series analysis methods for the electrome of common bean leaves, we identified that wound stimuli (W), heat shock (HS), and the combined stimuli (W+HS) caused local and rapid systemic reactions and that the pattern of dynamics observed for single stimuli is different from those applied in combination.

Frequencies with higher amplitude and more energy, as shown by the results of ABP and DFA exponent (Figure 5, Figure 6, Figure 7 and Figure 8), were found in the first stretches of the time series, followed by a reduction in ApEn values that indicates greater regularity, which may be associated with stress effects on dynamical aspects of plant physiology [14]. The studies by Souza et al. [36] and Saraiva et al. [14] evaluated the β exponent (an energy indicator of frequencies) after submitting soybean plants to different stress conditions increasing the exponent concerning the control treatment. Stressful conditions demand more energy from the system, and this energy may be behind the triggering of the systemic change in the electrome, considering that after the cut, the frequencies of local and systemic electromes showed an increase in the amplitude for W and W+HS, but at a lower intensity than that observed in local and systemic HS. Therefore, the need for frequencies with sufficient energy to trigger a consistent systemic change in electrome is apparent.

Differently from the isolated stimuli, the combination of stimuli (W+HS) promoted a biphasic behavior of the electrome, indicating a possible integration of the signals triggered by these stimuli. Similar behavior was observed when a combination of heat and re-irrigation was applied to maize plants, generating a heat-induced variation potential, followed by a transient depolarization induced by re-irrigation [37]. 

The results of turgor pressure variation in the systemic leaves indicated that a hydraulic signal accompanies the electrical signal in inducing a specific systemic response to stimuli, as well as changes in ROS levels. Among the changes in hydraulic dynamics, the most evident was observed after the HS and W+HS stimuli. In the study by Vuralhan-Eckert et al. [35], in which stomatal conductivity and photosynthesis were evaluated, the application of both stimuli at the same time showed that maize plants respond first to injury events and then to processes induced by re-irrigation. This observation is consistent with our results for turgor pressure variation, in which the hydraulic dynamics after W+HS present a pattern similar to W, but with greater amplitude and duration, possibly associated with the additive effect of the hydraulic waves induced by the combined stimulus.

When leaflets are damaged by burning or wound, there is an immediate loss of water content in the plants, and on many occasions, these injuries can disturb the plant’s vascular system, which has a direct effect on the turgor pressure of epidermal cells [37,38]. According to Johns et al. [5], physical damage that disrupts the integrity of the xylem should release tension in these vessels, and due to the relatively incompressible nature of water, this pressure change will be transmitted almost instantly through the vascular tissues. For this reason, it is possible to observe changes in hydraulic dynamics immediately after the stimuli.

Recently, a strong connection between REDOX metabolism and systemic signaling in plants during stress has been uncovered [39]. Studies have shown that ROS levels, the expression of different ROS scavenging enzymes, and the level of different antioxidants exhibit a unique pattern during combination stress that is different from that found to be induced by each of the different stresses applied separately [40]. Herein, an increase in hydrogen peroxide and superoxide content after W and HS were observed in both local and systemic leaves (Figure 1). Therefore, the stimuli triggered an increase in the production of ROS in the local leaf which, in turn, signaled the production of ROS in the systemic leaves.

Studies evaluating systemic stomatal responses in Arabidopsis have shown dependence on systemic reactive oxygen species (ROS) signals and include systemic stomatal closing responses to light or injury, as well as systemic opening responses to heat [1,26,40,41]. In soybean plants, systemic stomatal responses were also associated with a dependence on systemic ROS signaling [6,26]. It is possible to associate the results mentioned above with those observed in our study. The results showed an increase in ROS after W and an increase in turgor pressure which is related to stomatal closure, while for HS, it is possible to see a significant increase in ROS and a decrease in turgor pressure, often related to opening stomata. In effect, stomatal opening increases leaf transpiration rate and, consequently, decreases water pressure inside leaves, reducing the whole turgor pressure [42].

The reduction in the level of lipid peroxidation observed only for the combination of stimuli suggests a rapid activation of defense mechanisms that could be better clarified through the evaluation of enzymatic and non-enzymatic defense mechanisms, not explored in this study. According to Dvořák et al. [43], in general, increased ROS production caused by a plethora of environmental stimuli rapidly triggers antioxidant defense by several mechanisms, including retrograde signaling, transcriptional control, post-transcriptional regulation, post-translational redox modifications or phosphorylation, and protein–protein interactions.

Although stimulation caused changes in hydraulic and electrical dynamics for a while, the behavior of these signals returns to pre-stimulation values minutes after the stimulus ceased (Figure 2, Figure 3, Figure 4, Figure 5, Figure 6, Figure 7, Figure 8 and Figure 9). According to Kranner et al. [44], the response of a plant to stress will vary depending on the duration (short and long term) and severity of stress. “Lower stress events” can be partially compensated by acclimatization repair mechanisms, while severe or chronic stress events cause considerable damage and can lead to cell and tissue death. Herein, local stimuli for a short period (see Section 4) did not lead to a “collapse” of the plant cells, which allowed the plant to return to a non-stimulated condition minutes after the application of the stimuli.

It has not gone unnoticed that after the stimuli, the electrome behaved in a manner that previous studies suggest to be a state of attention in plants [16,45]. According to the hypothesis developed by Parise et al. [45], attention in plants is a disproportionate investment of energy in an activity or the perception of a stimulus or set of stimuli [46], and it could be observed through electromic analyses when there is a drop in the electrome complexity accompanied by an increase in the correlation of the signals and, likely, an increase in the energy of the electrome [45]. This is precisely what was observed here. After stimulation, there was a transient decrease in the ApEn in both local and systemic leaves, together with an increase in the correlation and ABP. According to the hypothesis, when a plant faces a challenge, the modules must synchronize their functioning to respond in coordination to the stimuli perceived. Since they will be working in concert, there will be a decrease in the complexity of the electrical signals, which will become more regular and predictable. At the same time, there will necessarily be an increase in the correlation of the signals and a likely increase in their energy. This is what is called plant attention [45]. This state is not expected to last long, but only until the problem is solved, or the actions needed to face it are completed. Examples of such actions would be the delivery of information to distant modules and/or the achievement of a new physiological state (acclimatization) when the signs of attention in the electrome would not be detectable anymore. The transient behavior of plant attention, emphasized by Parise et al. [45], was also observed in this study, where attention-related alterations in the electrome lasted for around 15 min. Therefore, this work corroborates the hypothesis of plant attention.

Furthermore, we have noticed some unexpected changes in bioelectrical activity in the control (non-stimulated) plants, although slight and transitory (Figure 2, Figure 3, Figure 4, Figure 5, Figure 6, Figure 7 and Figure 8). Because the control plants were in the same Faraday cage as stimulated ones, we cannot discard the possibility that volatile organic compounds (VOCs) could have been released from the stressed plants and somehow affected the behavior of non-stimulated ones. It is known that VOCs are powerful signaling molecules allowing plant–plant communication, especially under stressful situations [33]. In this vein, it would be instructive to carry out specific studies to test such a possibility, i.e., to test the hypothesis that plant–plant communication mediated by VOCs can trigger bioelectrical changes in plants. Such a hypothesis has already been suggested by Parise et al. [16] when the electrome of *Cuscuta racemosa* showed evidence of being affected by a distant host.

## 4. Materials and Methods

### 4.1. Plant Material and Growing Conditions

Plants for the experiments were obtained by germination of common bean seeds (*Phaseolus vulgaris* L. cv. IAC Netuno) sowed in Gerbox^®^ boxes lined with germitest paper moistened with 15 mL of distilled water. When the roots were 1 to 2 cm long, the seedlings were transplanted into 380 mL polystyrene pots (drilled in the base) containing 450 g of washed and sterilized sand and kept under a customized lighting system composed of LED lamps providing a photosynthetically active photon flux density (DFFFA) of approximately 350 µmol m^2^ s^−1^. The lighting system was connected to a timer that set the photoperiod at 14 h light and 10 h dark. Air humidity was maintained around 74% and the temperature at 25 ± 1 °C.

During the growth period, the plants were watered daily with distilled water (40 mL), and three times a week, they were supplemented with Hoagland and Arnon (20 mL) nutrient solution [47]. The plants were kept under these conditions until the third trifoliate leaf was fully expanded.

### 4.2. Experimental Design and Evaluation of the Dynamics of Systemic Responses Induced by Simple and Combined Stimuli

To test our hypothesis, three trials were conducted, where in each one, the stimuli (treatments) of heat shock (HS), wounding (W), and heat shock + wounding were applied locally (i.e., on single leaves). Then, the time series of the bioelectrical signals and the variation in the turgor pressure, in addition to the samples for the analysis of ROS content and lipid peroxidation, were evaluated as follows.

For the evaluation of single stimuli, the central leaflet of the second trifoliate leaf was selected for the local application of the HS and W treatments, while the first trifoliate leaf was not stimulated (systemic tissue) (Figure 10A). For the combined stimuli, the central leaflet of the third and second leaves was subjected simultaneously to heat and wounding stimuli (HS+W treatment), and the first trifoliate leave remained not stimulated (systemic tissue) (Figure 10B). The essays performed are detailed below.Essay I—Heat shock (HS): The heat shock stimulus (HS) was applied by placing a flame approximately 10 cm from the central leaflet for 20 s. Measurements in unstimulated (control) plants were also obtained. Leaf temperature was measured with an infrared camera (FLIR Systems) on local leaves before and after stimulation to have an idea of leaf temperature variation during the test. According to the measurements, after stimulation, the local leaf temperature increased by an average of ±23 °C (data not shown).Essay II—Wounding (W): With calibrated scissors, 2 cm cuts in the central leaflet of the second trifoliate leaf were made to induce systemic wound signaling and response.Essay III—Heat shock + Wounding (HS+W): The stimuli were applied simultaneously to different leaves of the same plant to induce a signaling and systemic response to the combined stimulus. The 2 cm cut was applied to the third leaflet and the application of thermal shock was to the second leaflet for 20 s.

In each HS, W, and HS+W essay, electrome measurements, turgor pressure variation, quantification of ROs, and lipid peroxidation were obtained, in addition to measurements in unstimulated plants (control).

### 4.3. Quantification of ROS and Lipid Peroxidation

The leaf samples were collected a few seconds after the application of the stimuli, in different plants than those that were being used for bioelectrical analysis since leaf detachment will cause extreme bioelectrical reactions. The samples were immediately weighted, frozen with liquid nitrogen, and kept in an ultra-freezer. 

The superoxide (O_2_^−^) content was determined according to Li et al. [48] in all experiments. For extraction, tissues (0.2 g) were ground in 65 mM phosphate buffer, pH 7.8, and centrifuged at 5000 *g* for 10 min. The supernatant was mixed with 65 mM phosphate buffer, pH 7.8, and 10 mM hydroxylamine hydrochloride, and placed at 25 °C for 20 min. Then, 17 mM sulfanilamide and 7 mM alfa-naphthylamine at a final concentration were added to the mixture. The absorbance of the solution at 530 nm was measured after incubation for 20 min at 25 °C. A standard curve with nitrite dioxide (NO_2_^−^) radical was used to calculate the rate of O_2_^−^ generation. 

The levels of hydrogen peroxide (H_2_O_2_) were determined according to Velikova et al. [49] in each experiment. Tissues (0.2 g) were powdered in 0.1% acid (w:v) trichloroacetic acid (TCA). The homogenate was centrifuged (12,000× *g*, 4 °C, 20 min), and the supernatant was added to 10 mM potassium phosphate buffer, pH 7.0, and 1 M potassium iodide. The absorbance of the reaction was measured at 390 nm. The H_2_O_2_ content was given on a standard curve prepared with known concentrations of H_2_O_2_. 

To measure lipid peroxidation, the thiobarbituric acid (TBA) test was used, which determines malondialdehyde (MDA) as the end product of lipid peroxidation [49]. The material (0.1 g) was homogenized in a 0.1% (w:v) TCA solution. The homogenate was centrifuged (12,000× *g*, 4 °C, 20 min), and the supernatant was added to 0.5% (w:v) TBA in a 10% TCA solution. The mixture was incubated in hot water (90 °C) for 20 min, and the reaction was stopped by placing the reaction tubes in an ice bath for 10 min. Then, the samples were centrifuged at 10,000× *g* for 5 min, and the absorbance was read at 535 nm. The value for non-specific absorption at 600 nm was subtracted. The amount of MDA-TBA complex (red pigment) was calculated from the extinction coefficient (ε = 155 × 10^3^ M^−1^ cm^−1^).

### 4.4. Electrome Acquisition and Analysis

#### 4.4.1. Electrophytogram (EPG)

Bioelectrical measurements were performed using the electrophytogram (EPG) method [14]. A Biopac Student Lab System was employed for bioelectrical data acquisition system, model MP-36 (Goleta, CA, USA), with 4 channels of high impedance (10 GΩ), SSL2 cables, and 12 mm thick stainless steel needle electrodes (model EL-452). A pair of electrodes, placed 1 cm from each other, was inserted in the plants to capture the signals from the local and systemic leaves. The measurements were performed in 1 plant not stimulated and 1 plant stimulated, simultaneously inside a Faraday cage to avoid electrostatic effects from the laboratory power net. In Essay III, one of the electrodes was introduced in the internode between stimulated leaves (sites I and II) and another in the petiole of the systemic leaf (not stimulated). 

The voltage variations (ΔV, measured in µV) captured by the electrodes go through a series of steps starting from the data acquisition system, amplification, filtering, and conversion of these signals. In this study, we used the electrocardiogram (ECG-AHA) function present in the MP-36’s BSL-PRO software [15,16]. We used amplifiers with high input impedance (>109 Ω) to avoid signal distortions. A minimum high-pass filter of 0.5 Hz and another low-pass filter of 1.5 kHz was used to filter the signal, so that all frequencies below 0.5 Hz and above 1.5 kHz were attenuated. In addition, a 60 Hz band-stop was adopted on all channels to avoid noise from electronic components present in the laboratory. In all experiments, a data capture frequency of 62.5 Hz with a gain (amplification) of 1000 times was used.

A period of 24 h of acclimatization to the electrodes was necessary before data acquisition since the moment of insertion of the electrodes causes wound responses in the plant that normalize after a few hours [50]. 

Electrome measurements of each plant were obtained in the form of time series of microvolt variation as Δ𝕍 = {ΔV1, ΔV2, …, ΔVn}, where ΔV is the potential difference between the electrodes inserted in the leaf petiole and n is the time series size. The duration of the series is derived from a one-hour sample of data acquisition, with an acquisition rate of 62.5 Hz, totaling n = 225,000 points.

The measurements of the local and systemic leaves were divided into two moments: (1) one hour of measurement before stimulation, and (2) one hour of measurement after stimulation. This form was used for each of the treatments. The W, HS, and W+HS treatments were classified as follows: tlb—treated local b; tla—treated local a; tsb—treated systemic b; tsa—treated systemic a. Control groups were classified as: clb—control local b; cla—control local a; csb—control systemic b; csa—control systemic a. Ninety time series (n = 225,000 points) were obtained in each group of the different treatments.

#### 4.4.2. Electrophysiological Analyzes

After acquiring the measurements, each time series was analyzed by the following techniques.

##### Visual Inspection

Visual inspection of time series allows a preliminary search for patterns or changes in the raw runs. Although descriptive and subjective, it allows a first analysis of the behavior of the time series, as well as some comparisons between them, such as the high presence of voltage variation peaks in specific stretches of the series [15,16].

##### Analysis of the Dispersion of Features over Time (Time Dispersion Analysis of Features—TDAF)

The TDAF method, developed by our research group, aims to demonstrate the dynamics of the electrome run characteristics (features). The latest studies published in the literature suggest that 3 min long measurements already contain enough information to identify behaviors in the plant electrome [16,17,18]. Thus, the TDAF assumes the use of the smallest possible unit of the analyzed series, considering the measurement equipment, and uses the dispersion theory [51] to analyze the dataset as a whole. Sometimes, analyses considering an entire time series (TS) can mask behaviors that happen within a few seconds or minutes. Thus, through several tests, we were able to observe the behavior of bioelectrical runs in different time frames. 

Considering this information, the first action in the generation of the features is the division (cut) of all the TSs of each treatment in a series of less than 1 min interchangeable, with a delay of 20%, aiming to eliminate tendencies that the TS may have in the cut point. For each group of treatments (W, HS, W+HS, and Control), a total of 9300 samples were obtained. The groups were classified as follows: tlb—treated local b; tla—treated local a; tsb—treated systemic b; tsa—treated systemic a; clb—local control b; cla—local control a; csb—systemic control b; csa—control systemic a.

In summary, the TDAF divides the series at the same instant in time for all the TSs analyzed. Afterward, each piece with its respective position in time receives this index (Figure 11). This marker is used to return all results to their respective bins. Thus, in the end, the bins will contain all the time series and features; with these values, it is possible to obtain the data dispersion (maximum/minimum values, median quartiles) minute by minute. 

An advantage of this analysis is the visualization of the behavior of the resources and how they differ, or not, for each class, without the need for any extra calculation, just manipulation via software. However, the major drawback is that it can only be used for time-bound event classifications, and highly diversified TS causes inaccurate and noisy analysis.

For each sample, we define the features based on different time series analysis techniques, which together help in electrome classification. 

##### Detrended Fluctuation Analysis (DFA) 

Detrended fluctuation analysis (DFA) calculates a power law scale estimate (similar to the Hurst exponent). For instance, by observing changes in the scale exponents, this method was able to identify heart diseases by analyzing the time series of an electrocardiogram [43]. Therefore, this analysis can calculate the autocorrelation of a time series and indicate its long-term and short-term memory index. The DFA calculation returns values that vary as follows: 1 − α < 1 the signal is stationary (oscillating around a constant mean) and can be modeled as fractional Gaussian noise; 2 − α = 0.5 indicates a lack of correlation or memory, indicating a noisy signal; 3 − 0.5 > α < 1 corresponds to a signal with positive correlation memory; 4 − α < 0.5 the correlation is negative; 5 − α > 1 the signal is non-stationary and can be modeled as fractional Brownian motion as H = a − 1 [52]. The Nolds library’s DFA method was used for resource generation. 

##### Average Band Power (Average Band Power—ABP) 

The ABP was calculated with the values obtained from the PSD. Briefly, this method analyzes the signal at specific frequencies such as “delta” (0.5–4 Hz), “theta” (4–8 Hz), “alpha” (8–12 Hz), “beta” (12–30 Hz), and “gamma” (30–100 Hz). It is commonly used to generate features in neuroscience EEG analysis [53,54,55,56,57,58,59]. To calculate the ABP, we integrated the PSD area determined by the region of interest. Composite Simpson’s rule was used, employing the Simpson method from the script library [60]. In general, Simpson’s rule decomposes the area into several parabolas and then sums them up, returning the total value of the area of interest. Herein, since the sampling rate was 62.5 Hz, the ABPs of the frequencies of 0–0.5 Hz (here called low {low}), 0.5–4 Hz delta, 4–8 Hz theta, 8–12 Hz alpha, and 12–30 Hz beta were calculated. 

##### Approximate Entropy 

Approximate entropy (ApEn) provides information about the level of organization of the time series and considers the temporal order of the points in the sequence of a time series; therefore, it is preferable to measure the regularity or randomness of biological signals [61,62]. Higher ApEn values indicate the existence of more irregular dynamics (greater complexity), while lower values indicate that the dynamics are more regular and deterministic [63,64,65]. 

The development of this analysis aimed to discriminate between two time series generated by different systems, or two time series generated by the same system under different physiological conditions [63,64,65]. 

Lower ApEn values were associated with compromised and deteriorated physiological processes; that is, they present greater regularity, while healthy physiological processes are more complex [61].

The definition of approximate entropy comes from Takens’ theorem [63], from which the set of M vectors Δvj = (Δvj, Δvj-τ, Δvj-2τ, …, Δvj-(m-1)τ) of m embedding dimension with lag τ, para j = 1, 2, 3, …, M and M = N − (m − 1)τ, where N is the size of the one-dimensional series (ΔV = ΔV1, ΔV2, ΔV3, …, ΔVN). For each vector j, the number of neighbors within a hypersphere of radius r is calculated. Subsequently, the mean of the logarithm of the number of neighbors is taken, which is given by (*r*)= 1M J-1 M*ln* (*N*(*j*)*viz* ≤ *r*). Thus, ApEn is defined in the following relation:*ApEn* = *Φm*(*r*) − *Φm* + 1(*r*),(1)

For the calculations, we set m = 2, τ = 1 e r = 0.02. σ, where σ is the standard deviation of the original time series. The choice of m = 2 is due to the better efficiency of the ApEn calculation for small time series [63].

In this study, the approximate entropy obtained from the time series was compared between the moments “before” and “after” each stimulus (single or combined).

### 4.5. Measurements of Turgor Pressure Variation 

The non-invasive Yara ZIM^®^ probe measures the pressure difference between two magnets and leaf turgor, called patch pressure (Pp), and therefore provides information on relative changes in leaf turgor in real time over long periods under laboratory or field conditions [66,67]. 

Here, we chose to install the probes only on the systemic leaf, due to the limited number of probes available, as well as difficulties during installation on the delicate leaves of common beans that sometimes led to tissue damage by the pressure exerted by the magnets. Furthermore, before the application of stimuli, we considered three days of observation after the probes’ installation to make sure that the installation was successful. After this period, measurements were carried out one hour before and one hour after the application of W, HS, and HS+W stimuli, with a frequency of one measurement every 10 s.

### 4.6. Statistical Analysis

ROS quantification and lipid peroxidation data were analyzed as a completely randomized design in a double factorial scheme, where one of the factors was the tissue (2 levels—local/systemic) and the treatments were the second factor (W, HS, and W+HS). Analysis of variance (ANOVA) was performed, followed by the Tukey test (*p* < 0.05), to assess the interaction between the factors. For electrical signal data, in the ABP, ApEn, and DFA analyses, the data are presented as median ± SD (standard deviation of the mean). The results for the turgor pressure variation measurements obtained by the Zim probe were calculated as the control percentage, which is the pressure measured on the leaf 1 h before the stimulus application. Each dataset includes SE of 6 biological repeats. For statistical analysis, the statistical programs RStudio 1.2.1335 and Sigmaplot 12.0 were used.

## 5. Conclusions

Our results support the hypothesis that ROS signals likely act together with hydraulic and electrical signals inducing systemic responses triggered by simple and combined local stimuli. However, although the transmission of electrical signals is important in the way plants respond to their environment, it is not yet known how plants deal with a situation in which external stimuli act simultaneously, nor how the different signals behave under these conditions. The present work, therefore, tries to take a step forward in understanding this interesting gap in plant ecophysiology. Our results showed that changes in electrome were different between types of isolated stimuli, including the combination of these, and that these electrical signals are accompanied by systemic changes in hydraulic and ROS dynamics. It is possible that the combination of stimuli in this study promoted an additive effect, inducing the activation of enzymatic and non-enzymatic antioxidant mechanisms more quickly, as well as inducing a more intense hydraulic wave and an electrical signature resulting from the signals triggered by each of the stimuli. 

In addition to the participation of electrical and hydraulic signals, associated with the responses in a variation of turgor pressure and ROS analyzed here, we do not rule out other systemic signals such as nitric oxide (NO), RNAs, hormones, and volatile organic compounds (VOCs) [68,69,70,71], acting together and helping to activate systemic responses. Indeed, a mix of systemic signals is supposed to coordinate a complex network of internal communication integrating stress responses throughout the whole plant [4,5,6], which could also be related to a state of attention in bean plants when perceiving and processing the stimuli [45,46].

Further studies are needed to address the interdependence of these signals. The combination of different stimuli, the application of inhibitors, or research with mutants indicate the influence of the electrical signal on the expression of a complex response. Likewise, the possible interference of plant–plant communication though volatiles or other means must be considered.

## Figures and Tables

**Figure 1 plants-12-00924-f001:**
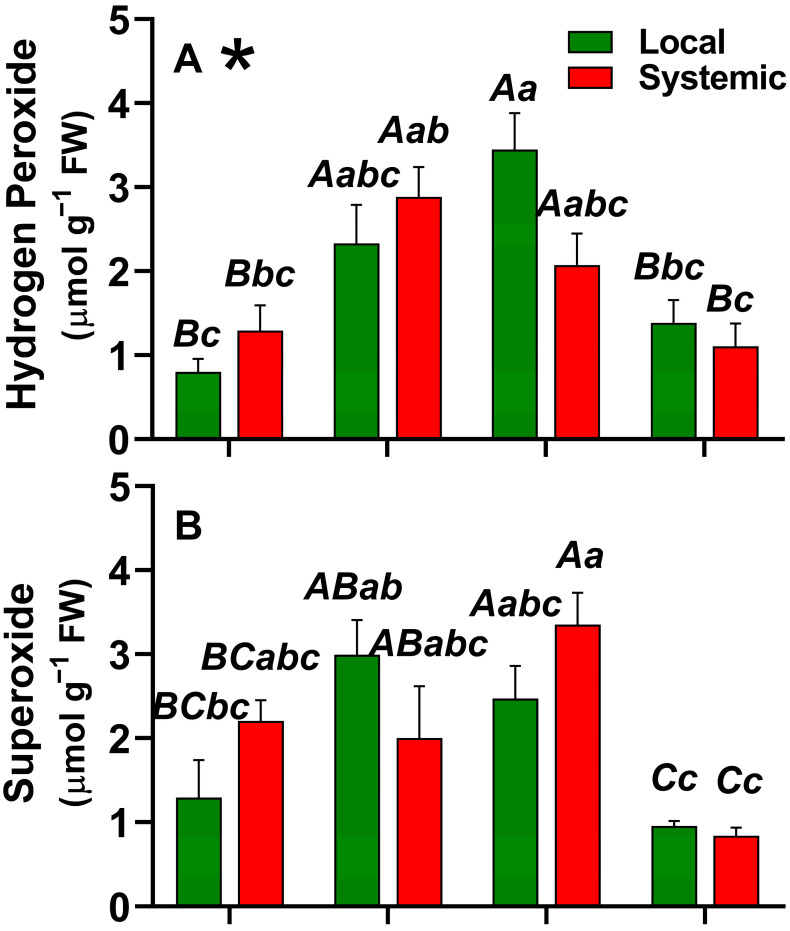
Hydrogen peroxide (**A**), superoxide (**B**), and lipid peroxidation (**C**) of local and systemic leaves submitted to different treatments. Capital letters represent differences between the treatment factor by the post hoc Tukey test, an asterisk next to the indicating letter in the figure indicates interaction between the factors through two-way ANOVA, and lowercase letters indicate the differences found by the Tukey test for the interaction between the factors of treatments and fabrics. C = control; W = wounding; HS = heat shock; W+HS = wounding + heat shock. *p* ≤ 0.05; n = 4.

**Figure 2 plants-12-00924-f002:**
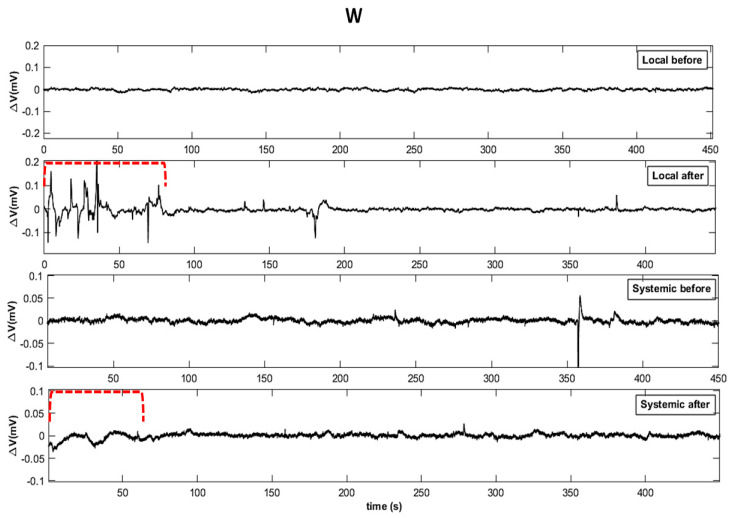
Original representation of the time series before and after the stimuli for wounding (W), heat shock (HS), and combined stimuli (W+HS) in local and systemic leaves. To facilitate comparison, the total length of the series (x = 1 h) was reduced to 7.5 min of measurement. The y-axis (ΔV) was adjusted for each situation. The red dotted line indicates the moment of greatest electrome change after stimulation.

**Figure 3 plants-12-00924-f003:**
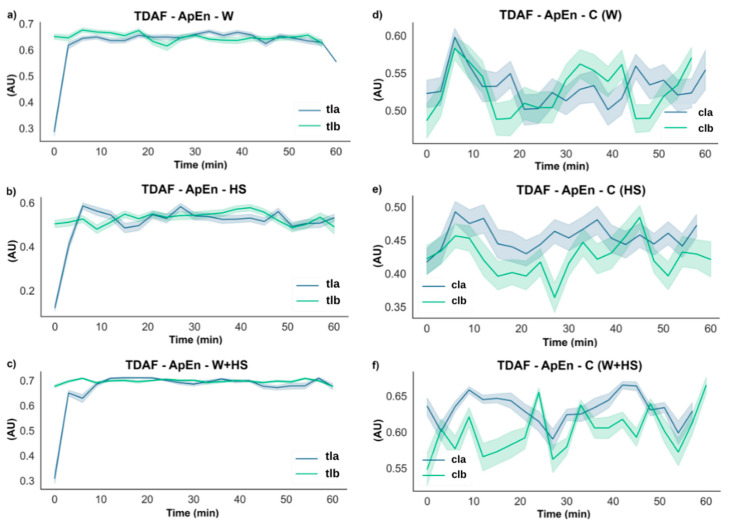
TDAF plots for ApEn analysis. Results for (**a**) the local wound (W), (**b**) heat shock (HS), and (**c**) wound + heat shock (W+HS). Results of the control plants for (**d**–**f**) wound C (W), heat shock C (HS), and wound + heat shock C (W+HS), respectively. tlb: treated local before stimulus; tla: treated local after stimulus; clb: control local before stimulus; cla: control local after stimulus. AU: arbitrary unit.

**Figure 4 plants-12-00924-f004:**
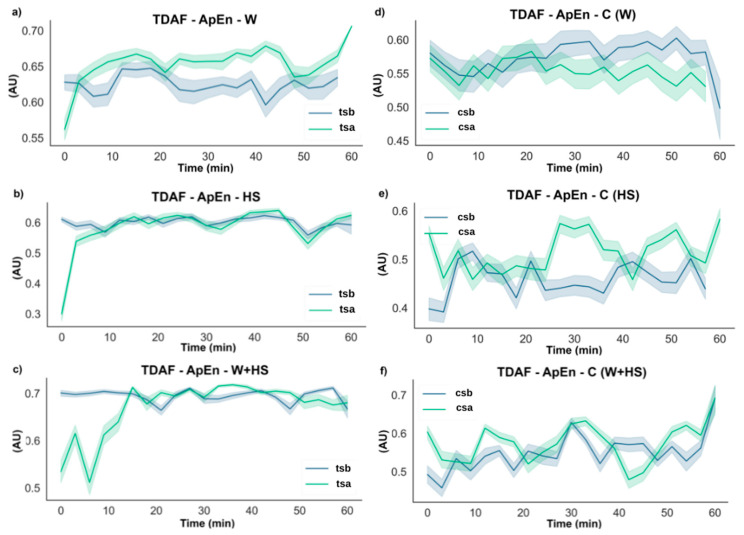
TDAF plots for ApEn analysis. Results for (**a**) systemic wounding (W), (**b**) heat shock (HS), and (**c**) wounding + heat shock (W+HS). Results of the control plants for (**d**–**f**) wounding C (W), heat shock C (HS), and wounding + heat shock C (W+HS), respectively. tsb: treated systemic before stimulus; tsa: treated systemic after stimulus; csb: control systemic before stimulus; csa: control systemic after stimulus. AU: arbitrary unit.

**Figure 5 plants-12-00924-f005:**
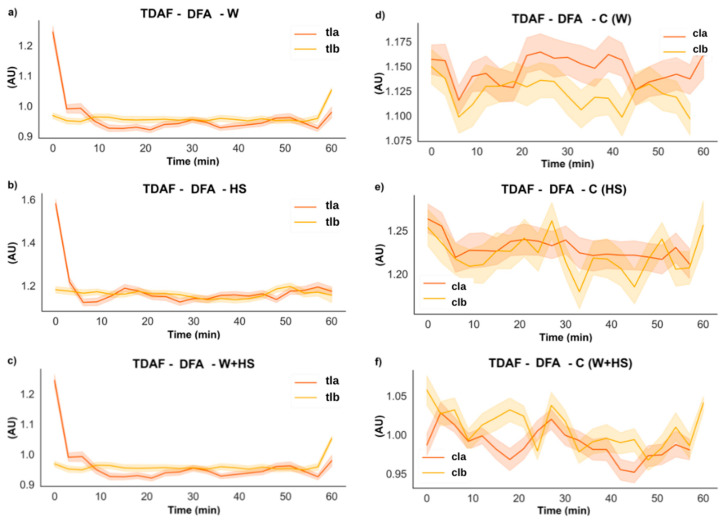
TDAF plots for DFA analysis. Results for (**a**) the local wound (W), (**b**) heat shock (HS), and (**c**) wound + heat shock (W+HS). Results of the control plants for (**d**–**f**) wound C (W), heat shock C (HS), and wound + heat shock C (W+HS), respectively. tlb: treated local before stimulus; tla: treated local after stimulus; clb: control local before stimulus; cla: control local after stimulus. AU: arbitrary unit.

**Figure 6 plants-12-00924-f006:**
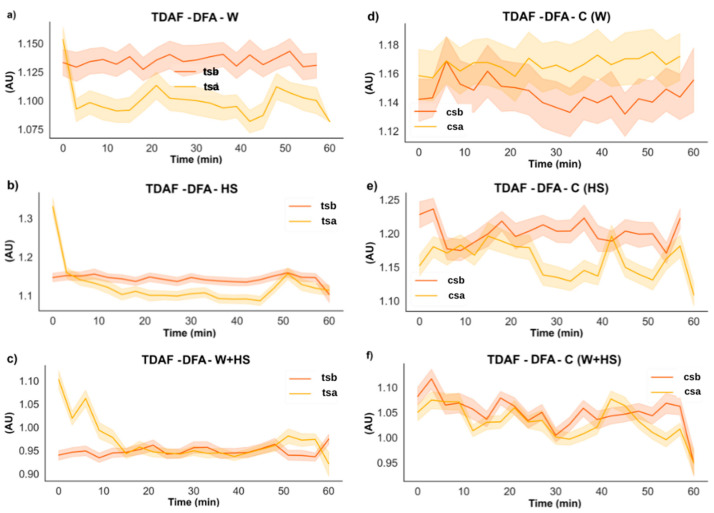
TDAF plots for DFA analysis. Results for (**a**) the systemic wound (W), (**b**) heat shock (HS), and (**c**) wound + heat shock (W+HS). Results of the control plants for (**d**–**f**) wound C (W), heat shock C (HS), and wound + heat shock C (W+HS), respectively. tsb: treated systemic before stimulus; tsa: treated systemic after stimulus; csb: control systemic before stimulus; csa: control systemic after stimulus. AU: arbitrary unit.

**Figure 7 plants-12-00924-f007:**
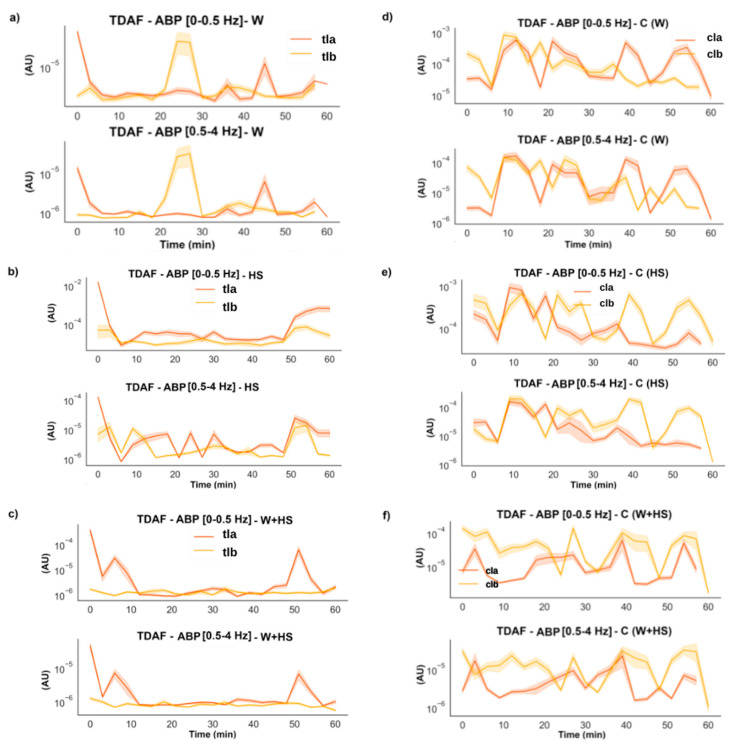
TDAF graphs for ABP analysis for low and Delta frequencies. Results for (**a**) the local wound (W), (**b**) heat shock (HS), and (**c**) wound + heat shock (W+HS). Results of the control plants for (**d**–**f**) wound C (W), heat shock C (HS), and wound + heat shock C (W+HS), respectively. tlb: treated local before stimulus; tla: treated local after stimulus; clb: control local before stimulus; cla: control local after stimulus. AU: arbitrary unit.

**Figure 8 plants-12-00924-f008:**
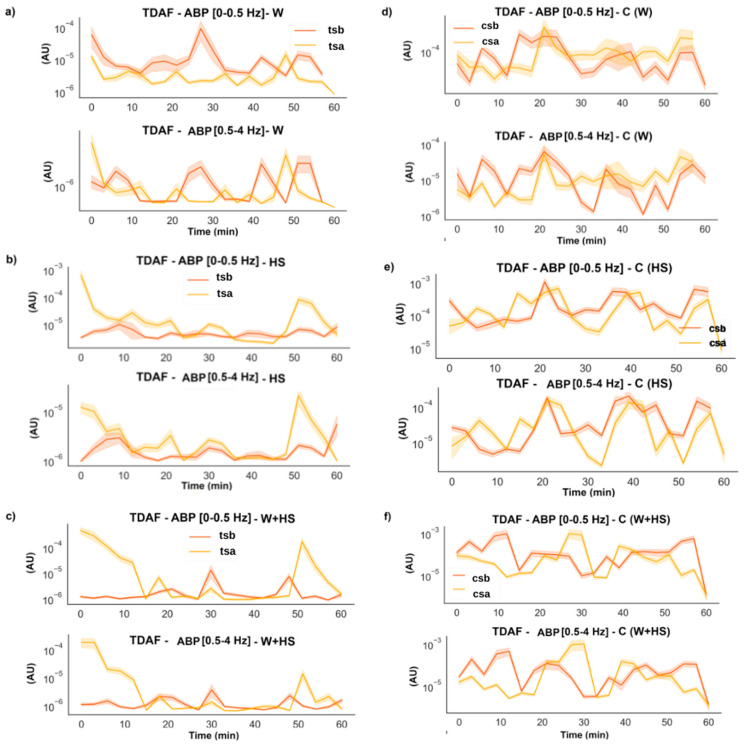
TDAF graphs for ABP analysis for low and delta frequencies. Results for (**a**) the systemic wound (W), (**b**) heat shock (HS), and (**c**) wound + heat shock (W+HS). Results of the control plants for (**d**–**f**) wound C (W), heat shock C (HS), and wound + heat shock C (W+HS), respectively. tsb: treated systemic before stimulus; tsa: treated systemic after stimulus; csb: control systemic before stimulus; csa: control systemic after stimulus. AU: arbitrary unit.

**Figure 9 plants-12-00924-f009:**
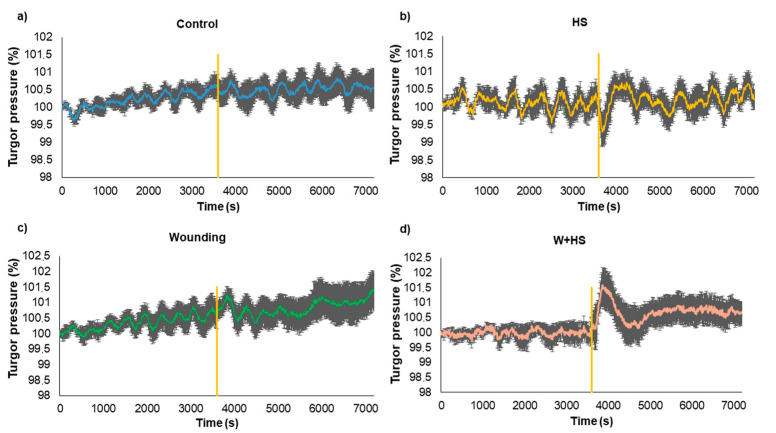
Measurements of the turgor pressure variation of systemic leaves over 2 h. Hydraulic pressure is represented as a percentage of the initial turgor pressure at 0 s. The stimuli (**a**–**d**) were applied after 1 h of measurement (represented by the vertical orange line). Each graph represents the average of 6 biological replicates (n = 6) and the standard deviation in each sampled point (time step = 10 s). Heat shock (HS), wound (W), and combined stimuli (W+HS). The average coefficients of variation are shown for the data before (C.V.B) and after (C.V.A) stimulation.

**Figure 10 plants-12-00924-f010:**
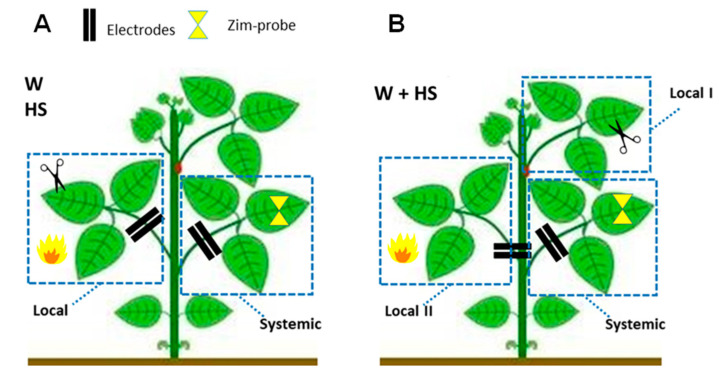
Representation of the experimental design indicating the place of application of the stimuli (W, HS, and W+HS) and where the systemic evaluation of the signals was performed. (**A**,**B**) The dashed rectangular border indicates the location (local tissue) where the different stimuli were applied. The yellow hourglass represents the turgor pressure probe installed on the systemic sheet. The two black lines indicate the electrode insertion region.

**Figure 11 plants-12-00924-f011:**
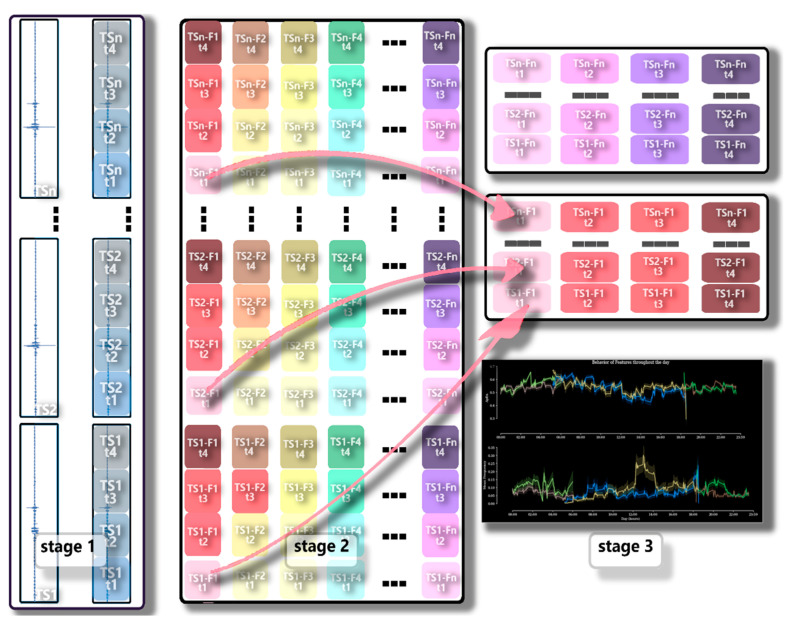
TDAF schematic. In stage 1, we have the time series (TS1, TS2, …, TSn) and the process of dividing and storing the position information referring to the time of each cut (t1, t2, t3, t4). In stage 2, the characteristics (F1, F2, F4, …, Fn) are calculated for each slice and together with the result, the timing information is virtually saved. In stage 3, time information is used for each generation and bin placement (data grouping). With each bin in place, the original time series is reconstructed, now with the results of each analyzed feature.

## Data Availability

Not applicable.

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
