# Peer review of "Systemic Signals Induced by Single and Combined Abiotic Stimuli in Common Bean Plants"

_plants, 2023, doi:10.3390/plants12040924_

Round 1

Reviewer 1 Report

Editor in Chief,

The manuscript titled “Systemic signals induced by single and combined abiotic stimuli in common bean plants” has been reviewed.

The authors provided several believable evidences to confirm the different effect of signals propagation stimulated by single and combined after wound and thermal shock treatments, measured by hydrogen peroxide, electrome dynamics and turgor pressure.

Totallthe content  could be acceptable with a few misses.

My question is Lack of units both of x and y axis from Fig 5 to Fig 10, and clear description about these Figs in the manuscript.

Author Response

Resp.: y axis from Fig 5 to 10 (now Figs. 3 to 8) are arbitrary units (AU), in x axis time is represented in minutes. Figures have been adjusted.

Reviewer 2 Report

Authors of the manuscript, “Plants-2147580”, performed a comparison analysis of the behavior of electrical (electrome), hydraulic signals, and combined stimuli in common bean plants against time. They used simple and mixed stimuli applications to identify biochemical responses and to extract information from the electrical and hydraulic patterns. In my opinion this research is interesting, and attempts to answer some of the questions in plant response to external stimuli. Nonetheless, I also believe the present MS version can be well-improved before it can be considered for publication. Below are listed few of my suggestions:

Major Comments

1.     The Introduction section is logically written; however, the English use in the whole document can be revised and paragraphing also improved.

2.     The result section can be briefly expanded; more result analysis can be done. Especially, Figure 5-10, there is more detail not explained.

3.     The Materials and Methods section is too lengthy, maybe some sections can be compressed into a single section and their information paraphrased.

Minor Comments

1.     In many parts, especially the Result section, the authors repeatedly use this sentence structure, …..it was found/observed + noun (Lines 179,184,185, 249). I strongly suggest to revise these parts.

2.     In-text referencing, the authors should refer to Author guidelines, Lines 293, 311 ..etc.

3.     Figure numbering is so confusing; the Figures can be numbered in order of appearance.

Author Response

Dear Reviewer,

Thank you very much for your considerations. We carried out the proper alteration in the new version of the ms.

Please, see below our point-by-point responses.

  1. The Introduction section is logically written; however, the English use in the whole document can be revised and paragraphing also improved.

Resp.: English and writing have been revised and improved throughout the manuscript.

  1. The result section can be briefly expanded; more result analysis can be done. Especially, Figure 5-10, there is more detail not explained.

Resp.: We have noticed a pattern in the changes of ApEn and ABP that was now properly explored in the ms.

  1. The Materials and Methods section is too lengthy, maybe some sections can be compressed into a single section and their information paraphrased.

Resp.: Changes were made to the materials and methods section to shorten its length.

Minor Comments

  1. In many parts, especially the Result section, the authors repeatedly use this sentence structure, …..it was found/observed + noun (Lines 179,184,185, 249). I strongly suggest to revise these parts.

Resp.: Suggested parts have been revised.

  1. In-text referencing, the authors should refer to Author guidelines, Lines 293, 311 ..etc.

Resp.: References in the text that presented a direct citation to the authors were rewritten according to examples from other articles in the same journal.

  1. Figure numbering is so confusing; the Figures can be numbered in order of appearance.

Resp.: Figures have been numbered in order of appearance, and their citations have been adjusted to match the new numbering.

Reviewer 3 Report

Very good paper, should be pubished as it is

Author Response

Thank you very much for having appreciate our work.